# Evidence-based intrapartum care practice and associated factors among obstetric care providers working in hospitals of the four Wollega Zones, Oromia, Ethiopia

Seid Wodajo[1]*, Alemnesh Mosisa[2], Dawit Misganaw[1], Ambaye Minayehu[1], Berhane Teklay[1], Yegoraw Gashaw[3], Yimenu Gardie[3], Yilkal Dagnaw[1], Adugna Olani[2]

1 Department of Midwifery, College of Health Sciences, Assosa University, Assosa, Ethiopia, 2 Department of Nursing, Institute of Health Sciences, Wollega University, Nekemte, Ethiopia, 3 Department of Nursing, College of Health Sciences, Assosa University, Assosa, Ethiopia

* seidwodajo7@gmail.com

**Data Availability Statement:** All relevant data are within the paper and its Supporting information files.

## Abstract

### Background

Even though Evidence-Based Practice (EBP) is a key component of quality of Intrapartum care and links to improved health care outcomes, consistent application of EBP in patient care remains a challenge for health care providers. In the study area, there are no previous studies conducted on evidence-based Intrapartum care practice among obstetric care providers. Therefore, this study was aimed to assess the magnitude of evidence based intrapartum care practice and its associated factors among obstetric care providers working in hospitals of Wollega zones, Oromia Region, West Ethiopia, 2022.

### Method

An institution-based cross-sectional study using quantitative method was conducted from January to April/2022 in 11 hospitals of the four Wollega zones. All obstetric care providers (278) who were practicing intrapartum care in the selected hospitals were included. The data was collected using structured self-administered questionnaire and observational checklist. Data was entered via Epi-Data version 3.1 and analyzed by SPSS version 25 statistical software. To see the association between the independent variables and evidence based Intrapartum care practice, multivariable logistic regression analysis was done. The statistical significance of association was declared at p-value ≤ 0.05. Tables, figures and charts were also used in descriptive statistics.

### Result

The overall magnitude of evidence-based Intrapartum care practice was found to be 63.7% [95% CI (59.7, 67.7)]. There was a statistically significant association between evidence-based Intrapartum care practice and having good knowledge about Intrapartum care practice [AOR = 2.95; 95% CI (1.52,5.73)], positive attitude towards Intrapartum care practice

**Funding:** The authors received no specific funding for this work.

**Competing interests:** The authors have declared that no competing interests exist.

[AOR = 3.13; 95% CI (1.59,6.16)], availability of updated Intrapartum care guideline [AOR = 2.88; 95% CI (1.46,5.70)], number of obstetric care providers per a shift ($\geq$5 care providers) [AOR = 2.31; 95% CI (1.01,5.29)], number of deliveries within a day (<10 deliveries) [AOR = 4.61; 95% CI (2.28,9.31)], educational level (MSc and above) [AOR = 5.75; 95% CI (2.23,14.84)] at p-value $\leq$ 0.05.

## Conclusion

Our study revealed that, magnitude of evidence-based Intrapartum care practice was found to be low according to the WHO recommendation. These findings indicate that additional attention and monitoring is required to implement current Intrapartum care practices with the WHO guidelines.

## Introduction

Evidence-based care is defined as the careful application of the best available evidence so that the clinician and the client can make the best decision possible, taking into account the client's requirements and values [1]. It requires three principles. The first is the care providers' ability to use correct information required under particular situation. The second requires the clinicians consider the needs and values of individuals receiving care. The third requires that the care providers take into account patient choices in the particular situation [2]. The application of these principles of evidence-based care practice helps to foster best care for women during intrapartum period. Evidence-based care (EBC) is therefore, a fundamental component of maternal and newborn health care which is an effective way to enhance intrapartum care quality [3] and to reduce maternal and neonatal death and morbidity [4,5].

In 2018, the World Health Organization (WHO) announced the most recent recommendations for intrapartum care which has four new classifications: recommended practices, not recommended, recommended only in specific contexts and recommended only in the context of rigorous research. Respectful maternity care, excellent communication, companionship throughout labor, and intermittent fetal heart rate auscultation are all examples of recommended practices. On the other hand, Some of the intrapartum care practices classified as not recommended practices like routine assessment of fetal well-being on labor admission, Perineal/pubic shaving, Continuous cardiotocography during labor, Routine amniotomy and Intravenous fluids for preventing labor delay [6].

Evidence-based care practice (EBCP) are very essential in providing the foundation for methods to increase the quality of intrapartum care [7] and has significant relevance to the birth outcomes [8]. So, integrating the latest evidence into childbirth practice can advance the health outcomes of childbearing women [9]. It is therefore, highly recommended to follow evidence based intrapartum care practice in order to reduce unacceptable variations in clinical practice such as by adhering to EBP guidelines, incorporating the most up-to-date research on EBP, using the WHO recommendations and by following care protocols [9–11].

Even though evidence-based practice (EBP) is a key component of quality of intrapartum care and links to improved health care outcomes, consistent application of EBP in patient care remains a challenge for health care providers [6]. Recent studies highlight limited application of generally accepted evidence-based clinical practices as well as limited application of more recent discoveries and recommendations. Many intrapartum care practices that are standard

policies in hospitals today were instituted in the 20th century and are still implemented without strong evidence for their effect on the laboring woman, labor progress, or newborn outcomes [12].

Globally, intrapartum care related complications are the leading cause of maternal and neonatal death often as a consequence of poor implementation of evidence based care practice. According to 2017 WHO estimation [13], everyday approximately 810 women died from causes related to poor implementation of intrapartum care practice in which Sub-Saharan Africa accounts for more than two-thirds of all these maternal deaths [14]. Global estimates of intrapartum related complications showed that intrapartum related neonatal deaths accounts for almost 10% of deaths in children aged under 5 years [15]. In Ethiopia, maternal mortality rate (MMR) of 412 per 100,000 live births [16] and a slight increase in neonatal mortality since 2016 from 29 to 33 deaths per 1,000 live births [17] is still too high with possible causes from intrapartum related complications such as hemorrhage, obstructed labour or sepsis.

## Methods and materials

### Study setting and period

The study was conducted from January 20 to April 20 /2022 in selected hospitals of the four Wollega Zones, Oromia, Ethiopia. These four independent zones namely: East Wollega, West Wollega, Kellem Wollega and Horo Gudru Wollega, are located in the western part of the country. The zonal capitals are Nekemte for East Wollega, Gimbi for West Wollega, Dembi Dollo for Kellem Wollega and Shambu for Horo Gudru Wollega. There are also other towns like Arjo, Nejo, Mendi, Sire, Gida Ayana, Kiremu and Gutin in the Zone.

In this research area there are two referral Hospitals (Wollega University Referral Hospital and Nekemte Specialized Hospital) located in East wollega zone, Nekemte town. The rest seventeen hospitals are Arjo, Sibu Sire, Gida, Shambu, Gudru, Abi Dengoro, Dembi Dollo, Hawa Gelan, Gidame, Kake, Gimbi, Begi, Nejo, Mendi, Bubo and two non-governmental hospitals (Aira and Gimbi Adventist) that are currently operational. Among those hospitals, Wollega University referral Hospital is the largest and the only teaching hospital, established in 2017 G. C. By now, this hospital gives service for the catchment area for about 8,000,000 populations and has 45 obstetric care providers. Nekemte Specialized Hospital is also another largest hospital, which is administered under the Federal Ministry of Health with 33 obstetric care providers at the time of study.

### Study design and population

An institution-based cross-sectional study design using quantitative method was employed. The source population comprised all obstetric care providers working in Hospitals of the four Wollega Zones. All obstetrics care providers who were available during the data collection period were the Study population.

Sample size determination and sampling procedures

Sample size determination

**Using single population proportion formula**:

$$n = \frac{(Z\alpha/2)\,2\,p(1-p)}{d^2}$$

Where,

n = the desired sample size,

p = population proportion,

Z = is the standard normal score

**Table 1. Summary of sample size determination using double population proportion formula to study evidence based intrapartum care practice and associated factors among obstetric care providers working in hospitals of Wollega zones, Oromia, Ethiopia, 2022.**

| Variables | Proportions | Sample Size | 10% non-response rate |
|---|---|---|---|
| Training on EBP | P1, (Exposed) = 45.5% | 128 | 141 |
| | P2, (Unexposed) = 70.1% | | |
| Health information | P1, Exposed) = 29.8%, | 300 | 330 |
| | P2, (Unexposed) = 45.5% | | |
| Knowledge | P1, (Exposed) = 53.7%, | 94 | 103 |
| | P2, (Unexposed) = 25.4% | | |
| Attitude | P1, (Exposed) = 58.3%, | 68 | 75 |
| | P2, (Unexposed) = 24.4% | | |

d = is the margin of error to be tolerated

$Z\alpha/2$ = Critical value for normal distribution

From similar study, the prevalence taken from the research done in Amhara regional state referral hospitals [18] were 38%. Hence, q = 1-p = 0.62, $Z\alpha/2$ = 1.96 (95% confidence level) and d = 4% (0.04).

Therefore,

$$n = \frac{(1.96)^2 (0.38)(0.62)}{(0.04)2} = 565.675 \approx 566$$

**Using double population proportion formula**:

By using Epi-Info 3.5.1 statistical package software considering two population proportion formula with the assumption using r = 1 (ratio of exposed and unexposed), 80% power, 95% confidence level ($Z_{\alpha/2}$ = 1.96), where P1 is the proportion of exposed and P2 is the proportion of unexposed. Different variables that are associated with evidence based intrapartum care were used.

The variable that yields highest sample size was selected. The variables used to calculate the sample size were training on EBP, Health information for teaching, knowledge and attitude taken from previous similar study [18]. From these four significant associated factors, we computed sample sizes 141, 330, 103 and 75 (all<566) (Table 1).

With the above inputs, the optimum sample size required for this study was 566. By taking 10% for non- response rate the final sample size became 623. However, since the total number of the population under investigation was small (278), Census method was applied (all Obstetric care providers in the study area included as study participants).

**Sampling procedure.** There are four zones and nineteen (19) functional hospitals in the study area. From the 19 hospitals, we selected eleven purposely (Wollega University Referral Hospital (WURH), Nekemte Specialized Hospital (NSH), Arjo Primary Hospital (ARH), Gimbi General Hospital (GGH), Nejo General Hospital (NGH), Mendi General Hospital (MDH), Aira General Hospital (AIH), Gimbi Adventist private Hospital (GAH), Shambu General Hospital (SHH), Gudru primary Hospital (GDH) and Dembi Dollo General Hospital (DWH). All Obstetric care providers working in the selected 11 hospitals (WURH = 45, NSH = 33, ARH = 18, GGH = 26, NGH = 25, MDH = 23, AIH = 24, GAH = 23, SHH = 22, GDH = 17, DWH = 22, Total = 278) were considered (S1 **Fig** in S1 File).

## Study variables

**Dependent variable**: Evidence-based intrapartum care practice

**Independent variables**:

**Socio-demographic related variables**: Age, sex, marital status, educational level, Profession and experience.

**Organizational related variables**: Access to computer, access to internet, availability of hospital guidelines, participation in professional activities and number of care providers per a shift and number of deliveries conducted within a day.

**Individual related variables**: Knowledge about intrapartum care practice, Professional attitude towards intrapartum care practice, searching for Scientific Journals, searching for databases and searching for WHO/Reproductive health library.

## Operational definition of variables

**Evidence-based intrapartum care practice**: It is a set of standard activities recommended by WHO and national protocols that are expected to be performed by obstetric care providers during intrapartum care. Obstetrics care providers who scored greater than or equal to the mean value of intrapartum-practice-related observational checklists [6,19,20].

**Intrapartum care**: refers to the care of a woman and her baby from the start of true labor through the first, second, third, and fourth stages of labor, which lasts one to two hours after the placenta is delivered [21,22].

**Knowledge**: Obstetric care providers who scored greater than or equal to the mean value of knowledge-related questions were considered as having a good knowledge while obstetric care providers who scored less than the mean value were considered as having poor knowledge [19,20].

**Attitude**: Those obstetric care providers who scored greater than or equal to the mean value of attitude related questions of intrapartum practice were labelled as having a positive attitude, whereas obstetric care providers who scored less than the mean values were considered as negative attitude [19,20].

## Data collection tools

Structured self-administered questionnaire and paper-based observational checklist was adapted from published articles and WHO recommendations on intra-partum care for a positive childbirth experience [6,18]. The self-administered questionnaire contained 35 questions arranged into four parts; socio-demographic, Organizational and individual related, knowledge towards intrapartum care and attitude towards intrapartum care related questions used to gather the required information. The observational checklist also contained 27 items of intrapartum care related questions used to measure the outcome variable, arranged into six parts: care throughout labor and birth, care in the 1st stage of labor, care during the 2nd stage of labor, care during the 3rd stage of labor, care of the newborn and care of the woman after birth, designed to elicit "**Yes**" or "**No**" response.

## Data collection procedures

Eighteen data collectors who are not staff members were recruited on the basis of being familiar with the same task before, and well known in the study area. The self-administered questionnaire was collected with the help of seven trained BSc midwives and five BSc midwife supervisors; whereas the observational data was collected using the observational checklist by eleven trained BSc Midwives.

## Data quality control

To ensure the quality of the data, the data collectors and supervisors had been appropriately trained for one day. At the institutions, data collectors were supervised daily by the supervisors and reported to the principal investigator on a daily basis. Filed questionnaires were checked daily for completeness and errors were corrected. Moreover, the self-administered questionnaire and observational checklist were pretested by taking 14 (5%) of the sample size two weeks prior to the main data collection time at Sibu Sire hospital, which is not included in the study. Corrections on the instrument, clarity, and ambiguity of words were made accordingly after the pretest was conducted.

## Data processing and analysis

The collected data had been checked for its completeness, and then it was coded and entered into Epi-Data version 3.1 and exported to the SPSS Version 25 statistical software package for cleaning and analysis. To explain the study population in relation to relevant variables, descriptive statistics, particularly frequencies, percentages, and descriptives like mean and standard deviation, were calculated. In addition, the cross-tabulation was computed using dependent and independent variables. To see the association between independent variable and dependent variable, bivariable and multivariable logistic regression analysis were carried out.

Those variables in bivariable logistic regression analysis whose p-value was less than 0.25 were included in multivariable logistic regression so as not to miss associated factors. Those crudely associated independent variables with the dependent variable using odds ratio with 95% confidence interval were entered via backward likelihood ratio for multivariable logistic regression analysis to adjust the influence of various independent variables (confounding effect) on the outcome variable. The model fitness was checked using Hosmer-Lemeshow goodness of fit test, which was (0.602), and multicollinearity test was checked by variable inflation factor (VIF) which was (1.106–1.202). The degree of association between independent and dependent variables was assessed by adjusted odds ratio using 95% confidence interval. Variables whose p-value ≤ 0.05 in multivariable logistic regression were considered as the cut-off point for statistically significant association. Finally, the result was written in the form of a text description, tables, and graphs.

## Ethical consideration and consent to participants

Ethical clearance and approval were obtained from the Wollega University research ethical review committee (WURERC), Institute of Health Sciences, Department of Nursing. The objectives of the study were attached to the questionnaire in detail, and then, written consent was taken from all study participants. Then, after obtaining informed consent from every participant, the data collectors continued the job by giving due respect to the norms, values, and beliefs of the study participants and ensuring the confidentiality of the data.

# Results

## Socio-demographic characteristics of study participants

A total of 278 respondents were participated in this study with a response rate of 100%. The study revealed that the mean age of the participants was 31.8 years (SD±4.08), ranging from 23–46. There was a male predominance 119 (63.7%) and 162 (58.3%) of the care providers were married. Majority, 233 (83.8%) of the respondents were Midwives followed by Medical Doctors 26 (9.4%), and over half of 168 (60.4%) had work experience of less than five year.

**Table 2. Socio-demographic characteristics of obstetric care providers working in hospitals of Wollega zones, Ethiopia, January 20 to April 20, 2022 (n = 278).**

| Variables | Category | Evidence-based intrapartum care practice | | Frequency (n = 278) | Percent (%) |
|---|---|---|---|---|---|
| | | Yes | No | | |
| Age | 23–27 | 19 (57.6%) | 14 (42.4%) | 33 | 11.9 |
| | 28–32 | 66 (48.9%) | 69 (51.1%) | 135 | 48.6 |
| | 33–37 | 66 (79.5%) | 17 (20.5%) | 83 | 29.9 |
| | ≥38 | 26 (96.3%) | 1 (3.7%) | 27 | 9.7 |
| Sex | Male | 119 (67.2%) | 58 (32.8%) | 177 | 63.7 |
| | Female | 58 (57.4%) | 43 (42.6%) | 101 | 36.3 |
| Marital status | Single | 65 (59.6%) | 44 (40.4%) | 109 | 39.2 |
| | Married | 109 (67.3%) | 53 (32.7%) | 162 | 58.3 |
| | Divorced | 3 (42.9%) | 4 (57.1%) | 7 | 2.5 |
| Profession | Medical Doctor | 25 (96.2%) | 1 (3.8%) | 26 | 9.4 |
| | Midwife | 134 (57.5%) | 99 (42.5%) | 233 | 83.8 |
| | IESO | 18 (94.7%) | 1 (5.3%) | 19 | 6.8 |
| Qualification | Senior | 11 (100%) | 0 | 11 | 4.0 |
| | GP | 6 (85.7%) | 1 (14.3%) | 7 | 2.5 |
| | Resident | 8 (100%) | 0 | 8 | 2.9 |
| | MSc | 38 (84.4%) | 7 (15.6%) | 45 | 16.2 |
| | BSc | 110 (55.3%) | 89 (44.7%) | 199 | 71.6 |
| | Diploma | 4 (50%) | 4 (50%) | 8 | 2.9 |
| Years of experience | <5 year | 88 (52.4%) | 80 (47.6%) | 168 | 60.4 |
| | ≥5 year | 89 (80.9%) | 21 (19.1%) | 110 | 39.6 |

Regarding the educational level of the study participants, majority 199 (71.6%) were BSc holders (Table 2).

## Magnitude of evidence-based intrapartum care practice

The magnitude of evidence-based intrapartum care practice (EBICP) was determined using 27 items of practice related questions, and the mean score of the respondent was 15.8 (SD±6). The overall magnitude of evidence based intrapartum care practice was 63.7% [95% CI (59.7, 67.7)]. From 278 observed care providers, 177 of them were applying EBICP (S2 **Fig** in S1 File).

From recommended practices, use of prophylactic Oxytocin (78.5%), applying controlled cord traction (77%), administration of prophylactic vitamin K (76.4%) and early initiation of breast feeding (75.3%) in the third stage of labor, use of techniques to prevent perineal trauma (73.5%) and encouraging urge to push in the expulsive phase (72.4%) in the second stage, and encouraging oral intake (71.6%) in the first stage were frequently performed practices in each stage of labor. While practices like administering opioid analgesia for labor pain relief (41.5%), using relaxation technique for labor pain management (53.8%), immediate post-partum maternal assessment (60%), encouraging maternal mobility (61.5%), digital vaginal examination every 4 hour (61.8%) were the less performed recommended intrapartum practices across each stage of labor.

Other beneficial obstetrical cares given throughout labor and birth like respectful maternity care, effective communication and companionship during labor and birth was also observed with magnitude of 65.8%, 63.2% and 64% respectively (S3 **Fig** in S1 File).

However, non-recommended intrapartum practices were also observed. Of these practices, sustained uterine massage (69.9%), routine nasal or oral suction (63.9%), routine amniotomy

**Table 3. Obstetric care providers level of knowledge and attitude based on their profession in hospitals of wollega zones, Ethiopia, January 20 to April 20, 2022 (n = 278).**

| Profession of the participant | Level of knowledge | | Level of attitude | |
|---|---|---|---|---|
| | Poor | Good | Negative | Positive |
| Medical Doctor | 5 (19.2%) | 21 (80.8%) | 10 (38.5%) | 16 (61.5%) |
| Midwife | 109 (46.8%) | 124 (53.2%) | 116 (49.8%) | 117 (50.2%) |
| IESO | 7 (36.8%) | 12 (63.2%) | 11 (57.9%) | 8 (42.1%) |
| Total | 121 (43.5%) | 157 (56.5%) | 137 (49.3%) | 141 (50.7%) |

(45%), routine antibiotic prophylaxis for uncomplicated vaginal birth (44.6%) were the commonly experienced procedures. Whereas, perineal/pubic shaving was not applied in majority of the cases (8.4%) (S4 **Fig** in S1 File).

## Knowledge and attitude of obstetric care providers about Evidence-based intrapartum care practice

The number of obstetric care providers who had good knowledge was 157 (56.5%), whereas the number of care providers with Positive attitude was 141 (50.7%). Medical Doctors showed good knowledge level (80.8%) followed by IESO (63.2%). Positive attitude towards evidence based intrapartum care was also more experienced by Medical Doctors (61.5%) followed by Midwives (50.2%) (Table 3).

This study also found that the study participants who knew evidence based intrapartum care for a positive childbirth care was 191 (68.7%), beneficial and non-beneficial obstetrical cares 202 (72.7%), effective intrapartum care throughout labor and birth 196 (70.5%) and effective immediate care of the newborn 198 (71.2%). Among 191 study participants who knew about evidence-based intrapartum care, majority 153 (80.1%) understand as it should be given based on WHO updated guidelines (S5 **Fig** in S1 File).

## Organizational and individual related factors

Of the total 278 study participants, only 105 (37.8%) had computer access at their work place, access to internet 107 (38.5%), access to updated intrapartum care guidelines 141 (50.7%), sources to access health information 237 (85.3%) and participation on professional activities related to their area of work in the last six months 210 (75.5%). Among those who had access to computer at their work place, more than half 65 (61.9%) were using computer for scientific reading and documentation equally. Participants utilized internet mainly for accessing online journals (62.6%), WHO Reproductive Health Library (40.2%) and Cochrane database (33.6%). On the other hand, among 237 respondents who had access to source of health information, majorities 190 (80.2%) obtained health information from books (S6 **Fig** in S1 File).

Regarding the number of obstetrics care providers and number of deliveries, more than half 213 (76.6%) respondents reported that average number of obstetric care providers per shift working in the labor/delivery unit was less than five. Similarly, majority 158 (56.8%) of respondents also reported that the average number of deliveries conducted within the day was ≥10 with mean of 10.2 (SD±4) ranging from 3 to 26 deliveries (S7 **Fig** in S1 File).

## Factors associated with evidence-based intrapartum care practice in a bivariable logistic regression

Sex of the respondent (female), level of education (MSc and above), obstetric care provider's work experience in labor and delivery (≥5 year) and having good knowledge about intrapartum

**Table 4. Factors associated with Evidence-based intrapartum Care Practice in bivariable logistic regression among obstetric care providers working in hospitals of wollega zones, Ethiopia, January 20 to April 20, 2022 (n = 278).**

| Variables | Categories | Evidence-based Intrapartum Care Practice | | COR (95% CI) | P-Value |
|---|---|---|---|---|---|
| | | No | Yes | | |
| Sex | Male | 58 | 119 | 1 | 1 |
| | Female | 43 | 58 | 0.65 (0.39,1.08) | 0.103 |
| Educational level | Below MSc | 93 | 114 | 1 | 1 |
| | MSc and above | 8 | 63 | 6.42 (2.93,14.08) | <0.001 |
| Work experience in labor and delivery | <5 year | 80 | 88 | 1 | 1 |
| | ≥5 year | 21 | 89 | 3.85 (2.19,6.77) | <0.001 |
| Knowledge towards intrapartum practice | Poor | 66 | 55 | 1 | 1 |
| | Good | 35 | 122 | 4.18 (2.48,7.02) | <0.001 |
| Attitude towards intrapartum practice | Negative | 71 | 66 | 1 | 1 |
| | Positive | 30 | 111 | 3.98 (2.35,6.72) | <0.001 |
| Availability of updated intrapartum care guideline | No | 64 | 73 | 1 | 1 |
| | Yes | 37 | 104 | 2.46 (1.49,4.07) | <0.001 |
| Use sources to access health information | No | 23 | 18 | 1 | 1 |
| | Yes | 78 | 159 | 2.6 (1.32,5.10) | 0.005 |
| Participated in professional activities in the last 6 months | No | 41 | 27 | 1 | 1 |
| | Yes | 60 | 150 | 3.79 (2.14,6.71) | <0.001 |
| Number of obstetric care providers per shift | < 5 | 89 | 124 | 1 | 1 |
| | ≥ 5 | 12 | 53 | 3.17 (1.60,6.27) | 0.001 |
| Number of delivery within a day | ≥ 10 | 80 | 78 | 1 | 1 |
| | < 10 | 21 | 99 | 4.83 (2.74,8.50) | <0.001 |

care had an association with evidence-based intrapartum care practice in bivariable logistic regression. Not only these, but also having positive attitude towards intrapartum practice, availability of updated intrapartum care guideline, having sources to access health information, Participation in professional activities, number of obstetric care providers per a shift (≥5 care providers) and number of deliveries within a day (<10 deliveries) were associated factors (Table 4).

**Significantly associated variables with evidence based Intrapartum care practice in a multivariable logistic regression.** Factors that were significantly associated with evidence-based intrapartum care practice were educational level (MSc and above), having good knowledge of intrapartum care, having positive attitude towards intrapartum care, availability of intrapartum care guidelines, number of obstetric care providers per shift (≥5 care providers) and number of deliveries conducted within a day (<10 deliveries) at P-value of ≤ 0.05.

The study revealed that the odds of providing evidence based intrapartum care practice in obstetrics care providers who had followed their postgraduate education was 5.8 times more likely [AOR = 5.75; 95 CI (2.23, 14.84)] when compared with BSc/diploma holders. In addition, the odds of providing evidence based intrapartum care practice was 2.9 times more likely in obstetric care providers who have intrapartum care guidelines compared with those who have not these guidelines in their working unit [AOR = 2.88; 95% CI (1.46, 5.70)].

In contrast to those OCPs with poor knowledge, the odds of providing evidence-based intrapartum practice was 2.9 times more likely in obstetric care providers having good knowledge of intrapartum care practice [AOR = 2.95; 95% CI (1.52, 5.73)]. Similarly, the odds of

**Table 5. Significantly associated variables with evidence-based intrapartum care practice in a multivariable logistic regression among obstetric care providers in hospitals of wollega zones, Ethiopia, January 20 to April 20, 2022 (n = 278).**

| Variables | Categories | Evidence-based intrapartum care practice | | AOR (95% CI) | P-Value |
|---|---|---|---|---|---|
| | | No | Yes | | |
| Educational level | Below MSc | 93 | 114 | 1 | 1 |
| | MSc and above | 8 | 63 | 5.75 (2.23,14.84) | <0.001 |
| Knowledge about intrapartum care practice | Poor | 66 | 55 | 1 | 1 |
| | Good | 35 | 122 | 2.95 (1.52,5.73) | 0.001 |
| Attitude towards intrapartum care practice | Negative | 71 | 66 | 1 | 1 |
| | Positive | 30 | 111 | 3.13 (1.59,6.16) | 0.001 |
| Availability of updated intrapartum care guidelines | No | 64 | 73 | 1 | 1 |
| | Yes | 37 | 104 | 2.88 (1.46,5.7) | 0.002 |
| Number of obstetric care providers per shift | < 5 | 89 | 124 | 1 | 1 |
| | ≥ 5 | 12 | 53 | 2.31 (1.01,5.29) | 0.047 |
| Number of deliveries within a day | ≥ 10 | 80 | 78 | 1 | 1 |
| | < 10 | 21 | 99 | 4.61 (2.28,9.31) | <0.001 |

**1** = Reference category, **COR** = Crude Odds Ratio, **AOR** = Adjusted odds ratio.

performing evidence-based intrapartum care practice was higher in obstetric care providers who have positive attitude towards intrapartum care practice [AOR = 3.13; 95% CI (1.59, 6.16)].

Higher number of obstetric care providers per shift was also positively associated with evidence-based intrapartum care practice compared to their counterparts. The odds of providing evidence-based intrapartum care practice with five and above care providers per a shift was 2.3 times more likely than those who were providing it with less than five care providers [AOR = 2.31; 95% CI (1.01, 5.29)]. Furthermore, number of deliveries conducted within a day was also another significant factor. The odds of performing intrapartum care practice based on the available evidence in those obstetric care providers who were conducting less than ten deliveries within a day were 4.6 times more likely compared with those who were conducting ten and above deliveries within a day [AOR = 4.61; 95% CI (2.28, 9.31)] (Table 5).

## Discussion of the findings

In this study, we have assessed magnitude of evidence-based intrapartum care practice and its associated factors. The finding revealed that the overall magnitude was 63.7%. This figure is higher than previous studies conducted in Ethiopia, in referral hospitals of Amhara regional state (38.2%) [18], in public hospitals of Addis Ababa (51%) [20] and in public hospitals of South Wollo zone (54.7%) [19]. This discrepancy might be due to a difference in the number of hospitals and sample size difference. Other reason might be due to difference in the study method, data collection tool and procedure might have a role.

However, this finding was lower than studies done in Lebanon (82%) [23] and Iran (78%) [24]. This might be due to cross-cultural difference, limited number of obstetric care providers serving in the hospital, provider's characteristics and additional duties other than obstetric care. The other reason may be due to variation of study setting. The finding of this study was also lower than study conducted in California (74.4%) [25]. This low performance of intrapartum care in current study may be due sample size difference, study area, data collection tool and study participants. Furthermore, some of the above studies were conducted in developed

countries, in which the service is given under advanced technology, high quality of health care institutions and due to difference in case over load.

In multivariable logistic regression, Having good knowledge about intrapartum care practice, positive attitude towards intrapartum care practice, availability of intrapartum care guidelines, Obstetric care providers with MSc and above qualification, higher number of obstetric care providers per a shift, lower number of delivery conducted within a day was positively associated with evidence-based intrapartum practice.

Obstetric care provider's (OCP's) knowledge and attitude were found to be strongly associated with evidence based intrapartum practice. OCPs who have good knowledge on intrapartum care practice were about three folds ahead to apply evidences during intrapartum care practice. This finding is consistent with a cross-sectional study conducted in Addis Ababa, Ethiopia [20]. It is also supported by most recent study conducted in south wollo zone of Ethiopia [19]. Another multi-institutional study done in United States also showed that health practitioners who had adequate knowledge was statistically associated with evidence based practice [26]. The explanation for this might be because individuals with sufficient knowledge would be more eager to put it into practice. This implies that acquiring adequate knowledge is a key for evidence based intrapartum practice. Similarly, having a positive attitude towards evidence based intrapartum care practice had nearly a triple fold increase in evidence based intrapartum practice. This finding also agrees with a study conducted in public hospitals of Addis Ababa, Ethiopia [20] and Eastern United States of America [27]. The reason for this could be because those with a positive mindset would be more eager to apply their expertise.

Additionally, educational level of the respondents had also strong association with evidence-based intrapartum care practice. OCPs having MSc and above qualification were nearly 6 times more likely to utilize evidences during intrapartum practice. This result is in agreement with cross-sectional studies done in other regions of Ethiopia [18,20] as the findings indicate that level of education positively affects performance of intrapartum practice. However, it does not agree with a study in done London [28]. The reason might be due to difference in the study population and setting.

Another important associated factor of evidence-based intrapartum care practice was availability of updated intrapartum guideline. Study participants who had access to updated intrapartum care guidelines at their work place were about 3 folds more likely to provide evidence based intrapartum care practice compared to their counterparts. This signifies that obstetric care providers who use intrapartum related guidelines to update their knowledge might have awareness how to apply latest best evidences in to clinical practice. In fact, daily reading habit of guidelines will help them to be motivated and put new evidence in to practice. OCPs who were more attached with reading BEmONC patient care manual, WHO safe child birth checklist and FMOH intrapartum care protocol had increased probability to use evidence in to practice, probably due to a motive to enhance maternal care.

Furthermore, this study also revealed that number of obstetric care providers per a shift and number of deliveries conducted within a day had significant association with evidence-based intrapartum care practice. The odds of providing evidence-based practice with five and above Obstetric care providers per a shift were 2.3 times more likely than those who were providing it with less than five Obstetric care providers per a shift. As evidenced by some studies [29,30] and by direct implication; the more OCPs assigned per shift, the more likely it is that they will perform efficiently. This therefore informs the need for public health authorities to draft appropriate measures that will ensure the recruitment of more obstetric care providers in public health facilities to improve quality of intrapartum care.

Evidence-based intrapartum care practice (EBICP) was also affected by number of deliveries. OCPs who were conducting less than ten deliveries per a day were about 4.6 times ahead to

provide intrapartum care practice based on the available evidence than who were conducting ten and above deliveries within a day. This finding is supported by studies done in Malawi [31]. This association may be due to the fact that the challenge and stress of being responsible for too large number of deliveries, or facing unmanageable workloads that exceed their capacity to cope has its own impact on the performance of the care providers. This finding will allow health care managers to assess the gap between current staffing levels and the levels required to meet a specific health facility's workload.

## Limitations

Our study was not ended without limitations, so this study has the following limitations. The potential for the Hawthorne effect could not be avoided due to the presence of an observer. The behavior of healthcare providers may have positively improved knowing that their practices were being observed. Therefore, the observed performance of recommended practices may be higher, and potentially harmful practices may be lower than in reality. The smaller study population compared to the actual calculated sample size was also another limitation.

## Conclusion and recommendations

The finding of this study showed that the proportion of evidence-based intrapartum practice among obstetric care providers working in Hospitals of the four wollega zone was low. Our study also found that non-recommended practices are still conducted frequently which is inconsistent with the WHO recommendations. Educational level (MSc and above), Having good knowledge on intrapartum care practice, positive attitude towards intrapartum care practice, availability of updated intrapartum guidelines, number of obstetric care providers per a shift ($\geq$5 care providers), number of deliveries within a day (<10 deliveries) were factors significantly associated with evidence based intrapartum care practice. These findings indicate that additional attention and monitoring are required to implement current intrapartum care practices with the WHO guidelines.

## Supporting information

**S1 Checklist.**
(DOCX)

**S1 File.**
(DOCX)

**S1 Dataset.**
(SAV)

## Acknowledgments

The authors thank the data collectors, data collectors' supervisors and administrative staff of all hospitals for their cooperation to conduct this research. We also extend our deepest gratitude to all clinical staff of the 11 hospitals for their unreserved support. Without them this work would not be accomplished.

## Author Contributions

**Conceptualization:** Seid Wodajo, Alemnesh Mosisa, Adugna Olani.

**Data curation:** Seid Wodajo, Alemnesh Mosisa, Dawit Misganaw, Adugna Olani.

**Formal analysis:** Seid Wodajo, Alemnesh Mosisa, Ambaye Minayehu, Berhane Teklay, Yegoraw Gashaw, Yimenu Gardie, Yilkal Dagnaw, Adugna Olani.

**Investigation:** Seid Wodajo, Alemnesh Mosisa, Dawit Misganaw, Ambaye Minayehu, Berhane Teklay, Yegoraw Gashaw, Yimenu Gardie, Yilkal Dagnaw, Adugna Olani.

**Methodology:** Seid Wodajo, Alemnesh Mosisa, Dawit Misganaw, Ambaye Minayehu, Berhane Teklay, Yegoraw Gashaw, Yimenu Gardie, Yilkal Dagnaw, Adugna Olani.

**Project administration:** Seid Wodajo, Adugna Olani.

**Resources:** Seid Wodajo.

**Software:** Seid Wodajo, Alemnesh Mosisa, Ambaye Minayehu, Berhane Teklay, Yegoraw Gashaw, Yimenu Gardie, Yilkal Dagnaw, Adugna Olani.

**Supervision:** Seid Wodajo, Alemnesh Mosisa, Dawit Misganaw, Ambaye Minayehu, Berhane Teklay, Yegoraw Gashaw, Yimenu Gardie, Adugna Olani.

**Validation:** Seid Wodajo, Alemnesh Mosisa, Dawit Misganaw, Berhane Teklay, Yegoraw Gashaw, Yimenu Gardie, Yilkal Dagnaw, Adugna Olani.

**Visualization:** Seid Wodajo, Alemnesh Mosisa, Dawit Misganaw, Ambaye Minayehu, Berhane Teklay, Yimenu Gardie, Adugna Olani.

**Writing – original draft:** Seid Wodajo.

**Writing – review & editing:** Seid Wodajo, Alemnesh Mosisa, Dawit Misganaw, Ambaye Minayehu, Berhane Teklay, Yegoraw Gashaw, Yimenu Gardie, Yilkal Dagnaw, Adugna Olani.

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
