## [Decision Letter · Decision Letter 0]

28 Nov 2022

PONE-D-22-25901Evidence-based intrapartum care practice and its associated factors among obstetric care providers working in hospitals of the four Wollega Zones, Oromia Region, West Ethiopia, 2022.PLOS ONE

Dear Dr. Wodajo,

Thank you for submitting your manuscript to PLOS ONE. After careful consideration, we feel that it has merit but does not fully meet PLOS ONE’s publication criteria as it currently stands. Therefore, we invite you to submit a revised version of the manuscript that addresses the points raised during the review process.

We look forward to receiving your revised manuscript.

Kind regards,

Kehinde Sharafadeen Okunade

Academic Editor

PLOS ONE

Journal Requirements:

Reviewers' comments:

Reviewer's Responses to Questions

**Comments to the Author**

1. Is the manuscript technically sound, and do the data support the conclusions?

Reviewer #1: Yes

Reviewer #2: Yes

2. Has the statistical analysis been performed appropriately and rigorously? 

Reviewer #1: Yes

Reviewer #2: Yes

3. Have the authors made all data underlying the findings in their manuscript fully available?

Reviewer #1: Yes

Reviewer #2: Yes

4. Is the manuscript presented in an intelligible fashion and written in standard English?

Reviewer #1: Yes

Reviewer #2: Yes

5. Review Comments to the Author

Reviewer #1: The manuscript was a questionnaire based survey conducted on health care workers and it was on their knowledge and practices of the World Health Organization evidence based practices of intrapartum care of women, within Healthcare facilities in Ethiopia. The work has been well written and it is a relevant aspect in Obstetric care.

I will kindly make the following comments to the Authors;

The introduction was quite long and it will be good if it can be reasonably and more concisely presented. The Authors should take note of typographical and text edit errors and advice on use of plagiarism checker for example, "While

Some of the intrapartum care classified as not recommended practices includes routine

assessment of fetal well-being on labor admission, Perineal/pubic shaving, Continuous

cardiotocography during labor, Routine amniotomy and Intravenous fluids for preventing

labor delay (8)." (page 10, line 1). The referencing method was haphazardly done and uniform pattern will be appreciated.

The P-value of <0.05, had been stated as being statistically for the determination of statistical significance for their data but it was noted that values above this (0.25), had been used for inclusion of variables in the multivariate analysis. This should be reviewed by the Authors to reduce bias. The service of a statistician can be helpful.

In the results section both texts and tables should not be used to present the results to reduce duplication of information.

The conclusion was also too long and should be be abridged.

The references should also be updated, as over 10% of the references appeared old.

The manuscript was well done and if these comments are noted and revised, the work is publishable.

Thank you.

Reviewer #2: Comments to Authors: The manuscript is well written is simple English language that was easy to understand. The topic is of contemporary relevance that will add to the body of knowledge. However, it requires minor revision in view of the points highlighted below:

The running title is too long and may be shortened as suggested in the manuscript

The detail of statistical analysis need not appear in the abstract

On page 17, the third sentence in results under socio-demographic characteristics needs modification thus: ‘I suggest that this should be reported in the context of positive Evidence-based intrapartum care practice and not overall respondent’.

On page 21 under ‘Factors associated with Evidence-based intrapartum care practice in a bivariable logistic regression’: The sentence is too long. Please break them into two for ease of understanding.

On page 25, the second paragraph the Authors should try and harmonize the possible reasons behind their lower findings compared to earlier studies.

On page 26, second paragraph, the highlighted sentence should be replaced with ‘was in contrast with the findings of Walsh D in London (32)’.

On page 27, under limitation: There was no mention of the smaller study population compared to the actual calculated sample size as a possible limitation

Concerning the second sentence on page 28, ‘While routine amniotomy was not a recommended practice according to WHO guidelines, is it really a harmful practice?’

On page 29, references 3 and 4 were repetitions

On same page 29, reference 5 and 6 should be properly cited

6. PLOS authors have the option to publish the peer review history of their article (what does this mean?). If published, this will include your full peer review and any attached files.

Reviewer #1: **Yes: **Adaramoye Victoria Olawunmi

Reviewer #2: **Yes: **DR YUSUF A. OSHODI

Associate Professor of Obstetrics and Gynaecology,

Lagos State University,

College of Medicine / Teaching Hospital, Ikeja, Lagos. Nigeria

---

## [Author Response · Author response to Decision Letter 0]

9 Dec 2022

Dear reviewers, we hope now the manuscript is clear and more acceptable than the previous one. We have tried to present the manuscript in proper manner according to your comments. Finally, we would like to extend our deepest gratitude for your endeavor for the improvement of this manuscript.

Both my Reviewers and the academic editor assigned for me were excellent. Thank you Dr. YUSUF A. OSHODI, Adaramoye Victoria and Kehinde Sharafadeen Okunde for your smart suggestions and comments.

---

## [Editor Report · Decision Letter 1]

27 Dec 2022

Evidence-based intrapartum care practice and associated factors among obstetric care providers working in hospitals of the four Wollega Zones, Oromia, Ethiopia

PONE-D-22-25901R1

Dear Dr. Wodajo,

We’re pleased to inform you that your manuscript has been judged scientifically suitable for publication and will be formally accepted for publication once it meets all outstanding technical requirements.

Kind regards,

Kehinde Sharafadeen Okunade

Academic Editor

PLOS ONE

Additional Editor Comments (optional):

The authors have addressed all the reviewers comments and the manuscript is now suitable for publication
---

## [Editor Report · Acceptance letter]

16 Jan 2023

PONE-D-22-25901R1 

*Evidence-based intrapartum care practice and associated factors among obstetric care providers working in hospitals of the four Wollega Zones, Oromia, Ethiopia*

Dear Dr. Wodajo:

I'm pleased to inform you that your manuscript has been deemed suitable for publication in PLOS ONE. Congratulations! Your manuscript is now with our production department. 

Kind regards, 

on behalf of

Dr. Kehinde Sharafadeen Okunade 

Academic Editor

PLOS ONE